# The effect of health insurance coverage on antenatal care utilization in Cambodia: A secondary analysis of Cambodia Demographic and Health Survey 2021–2022

Samnang Um [1,2]*, Channnarong Phan[1], Leng Dany[3], Khun Veha[4], Soklim Pay[5], Darapheak Chau[1]

1 National Institute of Public Health, Phnom Penh, Cambodia, 2 Faculty of Social Science and Humanities, Royal University of Phnom Penh, Phnom Penh, Cambodia, 3 The Elite Angkor Clinic, Siem Reap, Cambodia, 4 Department of Payment Certification of National Payment Certification Agency, Phnom Penh, Cambodia, 5 Technical Department of National Pediatric Hospital, Phnom Penh, Cambodia

* umsamnang56@gmail.com

## Abstract

Health insurance is essential in reducing or eliminating the financial constraint to accessing maternal health services caused by out-of-pocket payments. Also, it has a beneficial effect in minimizing maternal and child mortality. However, limited studies in Cambodia examined the association between health insurance coverage on antenatal care (ANC) utilization. Therefore, this study has examined the effect of health insurance coverage on ANC utilization in Cambodia. We utilized data from the 2021–2022 Cambodia Demographic and Health Surveys (CDHS), analyzing a total sample of 3,162 weighted women who gave birth within two years. Multiple logistic regression model using STATA V17 to assess the association between health insurance coverage with women who attended four or more ANC visits. About 24.9% of the women had health insurance coverage during 2021–2022. Most (86.1%) of women attended four or more ANC visits. Women with health insurance coverage were statistically significantly associated with attending four or more ANC visits with an adjusted odds ratio (AOR = 1.6; 95% CI: 1.1–2.4). Other factors significantly associated with attending four or more ANC visits include women with higher education (AOR = 3.1; 95% CI: 1.2–7.7), secondary education (AOR = 2.3; 95% CI: 1.5–3.5), richest households (AOR = 3.2; 95% CI: 1.5–6.8), and richer households (AOR = 1.9; 95% CI: 1.2–2.8). Pregnant women with health insurance coverage who had completed at least secondary education and had a better wealth index were more likely to attend at least four ANC visits. Thus, providing health insurance coverage and improving women's economic and educational may be essential to improving women's access to maternal health services in Cambodia.

## Introduction

Cambodia's maternal mortality rate has significantly declined in the past decade. Data from the 2021–2022 Cambodia Demographic and Health Surveys (CDHS) show that maternal

**Data Availability Statement:** The Cambodia Demographic and Health Survey data are publicly

available from the website: (URL: https://www.dhsprogram.com/data/dataset_admin).

**Funding:** The authors received no specific funding for this work.

**Competing interests:** The authors have declared that no competing interests exist.

mortality had declined dramatically, from 488 to 154 per 100,000 live births between 2000 and 2021–2022 [1,2]. By 2030, the global maternal death ratio is expected to drop to less than 70 per 100,000 live births, according to Sustainable Development Goals (SDGs) 3.1 [3]. This achievement can be attributed to the country's concerted effort to increase women's access to maternal health services, particularly the initiative to increase institutional births [4]. Institutional births dramatically increased, from 19.3% to 98%, while the proportion of pregnant women attending four or more antenatal care (ANC) appointments increased considerably, from 9% to 86.1%, between 2000 and 2021–2022 [5]. In several studies, women who had health insurance had higher rates of using maternal health treatments, such as timely ANC and attending four or more ANC visits [6–9].

In 2019, the total population of Cambodia was 15.55 million, with 17.8% living below the national poverty line [10]. Since 2016, Cambodia has been classified as a lower-middle-income country, with gross domestic product (GDP) per capita from 302 US dollars in 2000 to 1,625 US dollars in 2021 [11]. Also, current expenditures on health per capita significantly increased from 20 US dollars in 2000 to 116 US dollars in 2020 [12]. Globally, 50% of people cannot access essential health services, as the World Bank and World Health Organization (WHO) reported in 2017 [13].

Cambodian National Social Security Fund (NSSF) has provided health insurance coverage to formal sector workers [14]. And poor households are covered by the Health Equity Fund (HEF), the co-financing mechanism of the government and development partners [15]. By 2025, the government intends to expand the reach of the NSSF health insurance program to include the entire population [16]. Data from CDHS 2021–2022 indicated that 22% of women and 13% of men aged 15–49 years have any health insurance, respectively [5]. Health insurance coverage is expected to provide financial risk protection and reduce disparities in access by facilitating greater uptake of maternal health services [13]. To our knowledge, limited published peer-reviewed studies assess the association between health insurance coverage and access to maternal health services among women of reproductive age in Cambodia using updated data. One prior study on health insurance coverage and its impact on maternal health-care utilization in low- and middle-income countries utilized data from CDHS 2010 [8]. This study included all women and men aged 15–49 and pooled Demographic and Health Survey (DHS) data in 30 low-and middle-income countries (LMICs) [8]. An additional study aimed to assess levels of health insurance coverage in 30 LMICs and examines the impact of health insurance status on the use of maternal health care in eight countries spanning sub-Saharan Africa (Burundi et al., Namibia, and Rwanda), West Asia (Albania), and South and Southeast Asia (Cambodia and Indonesia) [8]. Several pieces of evidence on the effect of health insurance coverage on ANC utilization have been published [6–9,17–19]. The results indicate that women with health insurance coverage had higher odds of attending four or more ANC visits than those without health insurance coverage [6–9,17–19]. Moreover, those women who reported exposure to media, married women, those with high education, those living in wealthy economic families, those who are unemployed, and those living in urban areas were more likely to attend four or more ANC visits [6–9,17–19]. Given the limited study addressing this health concern among Cambodian women aged 15–49, we examined the effects of health insurance coverage on ANC utilization among women who had a live birth in the past two years in Cambodia. The findings will provide a broader perspective on levels of health insurance coverage and the impact of health insurance status on the use of maternal health care in Cambodia. Additionally, the study will enable policymakers to understand health insurance coverage among the adult population and proffer suggestions for improving universal health coverage in Cambodia.

## Material and methods

### Ethical statement

The CDHS 2021–2022 is publicly available, with all personal identifiers of study participants removed. Permission to analyze the data was granted by registering with the DHS program website at (URL: https://dhsprogram.com/data/available-datasets.cfm). Written informed consent was obtained from the parent/guardian of each participant under 18 before data collection. The Cambodia National Ethics Committee for Human Health Research (NECHR) approved the data collection tools and procedures for CHDS 2021–2022 for Health Research on 10 May 2021 (Reference number: 83 NECHR), and ICF's Institutional Review Board (IRB) in Rockville, Maryland, USA.

### Data source

We used data from the most recent CDHS (2021–2022), a household survey conducted every five years nationally representative of the population [5]. The two-stage stratified cluster sampling method collected the samples from all provinces. At the first stage, clusters, or enumeration areas (EAs) that represent the entire country (urban and rural), are randomly selected from the sampling frame using probability proportional (PPS) to cluster size. In the second stage, a complete listing of households was selected from each cluster using an equal probability systematic sampling. Then, interviews were conducted with women aged 15–49 years who were born in the five years preceding the survey in the complete list of selected households [5]. In total, 19,496 women aged 15–49 who had given birth in the last five years were interviewed face-to-face, using the survey standard questionnaire to collect information from women on several health indicators such as maternal health care service utilization, maternal and child health, nutrition, and reproductive health services [5]. Overall, 15,046 women who had not given birth in the past two years were excluded. Data restriction resulted in women who had a live birth in the past two years in a final analytic sample of 3,292 women (3,162 **weighted women**).

### Measurements

**Outcome variable.** This study's outcome was the number of ANC visits during the last pregnancy among women aged 15–49 years (coded as **0 = less than 4 ANC visits**, including women who reported no ANC visits, and **1 = four or more ANC visits**) [6,18,20].

**Independent variables.** The primary independent variable is maternal health insurance coverage (coded as 0 = no (reference and 1 = yes), including public and private insurance. The confounding variables included maternal factors: Women's age in years (coded as 1 = 15–30 (reference) and 2 = 31–49), marital status (coded as 1 = married (reference) and 2 = not married), birth order (coded as 1 = 1 (reference), 2 = 2–3, and 3 = 4 or more), education (coded as 0 = no education (reference), 1 = primary, and 2 = secondary or higher), occupation (coded as 0 = not working (reference), 1 = professional, 2 = sales or services, 3 = agricultural, and 4 = manual labor). Individual household factors, including the household wealth index (coded as 1 = poorest (reference), 2 = poorest, 3 = medium, 4 = richer, and 5 = richest), were calculated following the principal component analysis (PCA) [5]. Cambodia's geographical regions were grouped into four categories (coded as 1 = Plains (reference), 2 = Tonle Sap, 3 = Coastal/ Sea, and 4 = Mountains), and place of residences (coded as 1 = urban (reference) and 2 = rural) was defined based on Cambodia's General Population Census 2019 and adapted from the original CDHS 2021–2022 [5,10].

**Statistical analysis.** Statistical analysis was performed using STATA version 17 (Stata-Corp LLC). We applied for the DHS standard sampling weight variable (**v005/1,000,000**). Then, we used the survey-specific STATA command "**svy**" for descriptive and analytical analysis. Women's socio-economic and demographic characteristics were described using weighted frequency and percentage distributions.

Bivariate analysis using Chi-square tests assessed the association between the variables of interest (maternal and individual household characteristics) and ANC visits. All independent variables associated with ANC use at p-value ≤ 0.10 or that had a potential confounder variable [6,18] were included in the multiple logistic regression analysis to determine the independent factors related to ANC use [26]. Multicollinearity between original independent variables was checked, including women's age, number of children ever born, education, wealth index, occupation, marital status, health insurance coverage, and place of residence. The result of the evaluating variance inflation factor (VIF) scores after fitting an Ordinary Least Squares regression model with the mean value of VIF was 1.53, which is less than the cutoff point, indicating no collinearity correlation among the independent variables [27].

## Results

### Characteristics of the study population

Table 1 describes the socio-economic and demographic characteristics of the 3,162 women aged 15–49. The mean age was 22.2 years old (SD = 4.2 years); the age group of 15–29 years old accounted for 94.3%. The majority (95%) were currently married. More than 33.4% of women had their first child. Half of the women completed at least secondary education, while 10.6% had no formal education. Only 6.5% of workers were professionals, and 31.2% were unemployed. Of the sample, 20.7% of women were from the poorest households, and 19.7% were from poorer households. Sixty-two percent of the women lived in rural areas. Only 786 (24.9%) women aged 15–49 had health insurance coverage. 86.1% of women attended at least four ANC visits during pregnancy.

### Factors associated with four or more ANC visits in Chi-square analysis

In bivariate analysis (Table 2), a higher proportion of women with health insurance coverage had a significant association with four or more ANC visits (91.6% vs. 84.2%, p < 0.001). Women aged 31–49 reported being more likely to attend four or more ANC visits (88.0% vs. 86.0%, p < 0.001). Also, married women reported four or more ANC visits than nonmarried women (86.6% vs. 76.2%, p = 0.007). Women with no education were less likely to attend four or more ANC visits than those with higher education (71.0% vs. 95.6%, p < 0.001). Four or more ANC visits were higher among women working in professional (95.3%) and service (96.4%), respectively, compared to unemployed women (84.3%), with p < 0.001. Additionally, four or more ANC visits were higher among women from the richer and richest on the wealth index (94.8% and 89.4%, respectively), compared to the poorer and poorest (73.9% and 86.9%, respectively, with p < 0.001). Lastly, women living in urban areas reported higher four or more ANC visits than in rural areas (91.5% vs. 82.7%, p < 0.001).

### Association between health insurance and maternal healthcare services utilization

Table 3 shows the results of the multiple logistic regression analysis of the association between health insurance coverage and maternal healthcare services utilization after controlling for the socio-demographic factors. Compared to women without health insurance, those with health

**Table 1.  Socio-economic and demographic characteristics of women (N = 3,162 weighted).**

| Variables | | Freq. | % |
|---|---|---|---|
| **Mean age at the time of birth (SD)** | | 22.2(4.2) | |
| | 15–29 | 2,982 | 94.3 |
| | 30–49 | 180 | 5.7 |
| **Marital status** | | | |
| | Married | 3,004 | 95.0 |
| | Not married | 158 | 5.0 |
| **Birth order** | | | |
| | 1st child | 1055 | 33.4 |
| | 2nd or 3rd child | 1197 | 37.9 |
| | 4th child or higher | 910 | 28.7 |
| **Educational** | | | |
| | No education | 334 | 10.6 |
| | Primary | 1253 | 39.6 |
| | Secondary | 1361 | 43.0 |
| | Higher | 214 | 6.8 |
| **Occupation (N = 3,100)** | | | |
| | Not working | 986 | 31.2 |
| | Professional | 205 | 6.5 |
| | Sales | 579 | 18.3 |
| | Agricultural | 445 | 14.1 |
| | Services | 76 | 2.4 |
| | Manual labor | 810 | 25.6 |
| **Wealth index** | | | |
| | Poorest | 655 | 20.7 |
| | Poorer | 623 | 19.7 |
| | Middle | 626 | 19.8 |
| | Richer | 683 | 21.6 |
| | Richest | 574 | 18.2 |
| **Residence** | | | |
| | Urban | 1202 | 38.0 |
| | Rural | 1960 | 62.0 |
| **Region** | | | |
| | Plain | 1532 | 48.5 |
| | Tonle Sap | 996 | 31.5 |
| | Coastal | 201 | 6.4 |
| | Plateau/Mountain | 432 | 13.7 |
| **Covered by health insurance** | | | |
| | No | 2376 | 75.1 |
| | Yes | 786 | 24.9 |
| **Number of ANC visits** | | | |
| | < 4 ANC | 440 | 13.9 |
| | ≥ 4 ANC | 2722 | **86.1** |

*Notes*: Survey weights are applied to obtain weighted percentages. *****Plains**: *Phnom Penh, Kampong Cham, Tbong Khmum, Kandal, Prey Veng, Svay Rieng, and Takeo;* **Tonle Sap**: *Banteay Meanchey, Kampong Chhnang, Kampong Thom, Pursat, Siem Reap, Battambang, Pailin, and Otdar Meanchey;* **Coastal/sea**: *Kampot, Kep, Preah Sihanouk, and Koh Kong;* **Mountains**: *Kampong Speu, Kratie, Preah Vihear, Stung Treng, Mondul Kiri, and Ratanak Kiri.*

**Table 2. Maternal and household characteristics by women attending at least four antenatal care and delivery in a health facility (N = 3,162).**

| Variables | | Number of ANC visits | | p-value |
|---|---|---|---|---|
| | | Four or more | Less than four | |
| | | n = 2,722 | n = 440 | |
| | | % | % | |
| **Covered by health insurance** | | | | |
| | No | 84.2 | 15.8 | <0.001 |
| | Yes | 91.6 | 8.4 | |
| **Age at time of birth** | | | | |
| | 15–30 | 86.0 | 14.0 | <0.001 |
| | 31–49 | 88.0 | 12.0 | |
| **Marital status** | | | | |
| | Married | 86.6 | 13.4 | 0.007 |
| | Not married | 76.2 | 23.8 | |
| **Birth order** | | | | |
| | 1st child | 88.4 | 11.6 | <0.001 |
| | 2nd or 3rd child | 88.6 | 11.4 | |
| | 4th child or higher | 80.1 | 19.9 | |
| **Educational** | | | | |
| | No education | 71.0 | 29.0 | <0.001 |
| | Primary | 84.1 | 15.9 | |
| | Secondary | 90.1 | 9.9 | |
| | Higher | 95.6 | 4.4 | |
| **Occupation (N = 3,100)** | | | | |
| | Not working | 84.3 | 15.7 | <0.001 |
| | Professional | 95.3 | 4.7 | |
| | Sales | 84.4 | 15.6 | |
| | Agricultural | 79.0 | 21.0 | |
| | Services | 96.4 | 3.6 | |
| | Manual labor | 89.9 | 10.1 | |
| **Wealth index** | | | | |
| | Poorest | 73.9 | 26.1 | <0.001 |
| | Poorer | 86.9 | 13.1 | |
| | Middle | 86.3 | 13.7 | |
| | Richer | 89.4 | 10.6 | |
| | Richest | 94.8 | 5.2 | |
| **Residence** | | | | |
| | Urban | 91.5 | 8.5 | <0.001 |
| | Rural | 82.7 | 17.3 | |
| **Region** | | | | |
| | Plain | 89.3 | 10.7 | <0.001 |
| | Tonle Sap | 86.6 | 13.4 | |
| | Coastal | 89.0 | 11.0 | |
| | Plateau/Mountain | 72.2 | 27.8 | |

*Notes*: Survey weights are applied to obtain weighted percentages. *Plains: *Phnom Penh, Kampong Cham, Tbong Khmum, Kandal, Prey Veng, Svay Rieng, and Takeo; **Tonle Sap**: *Banteay Meanchey, Kampong Chhnang, Kampong Thom, Pursat, Siem Reap, Battambang, Pailin, and Otdar Meanchey;* **Coastal/sea**: *Kampot, Kep, Preah Sihanouk, and Koh Kong;* **Mountains**: *Kampong Speu, Kratie, Preah Vihear, Stung Treng, Mondul Kiri, and Ratanak Kiri.*

**Table 3. Association between health insurance and four or more ANC visits in simple and multiple logistic regression model.**

| Variables | | Four or more ANC visits | | | |
| --- | --- | --- | --- | --- | --- |
| | | Unadjusted (N = 3,162) | | Adjusted (N = 3,100) | |
| | | OR | 95% CI | AOR | 95% CI |
| **Covered by health insurance** | | | | | |
| | No | Ref. | | Ref. | |
| | Yes | 2.0*** | (1.4–2.9) | **1.6**[*] | **(1.1–2.4)** |
| **Age at time of birth** | | | | | |
| | 15–30 | Ref. | | Ref. | |
| | 31–49 | 1.2 | (0.7–2.1) | 1.1 | (0.6–2.0) |
| **Marital status** | | | | | |
| | Married | Ref. | | Ref. | |
| | Not married | 0.5** | (0.3–0.8) | **0.5**[**] | **(0.3–0.8)** |
| **Birth order** | | | | | |
| | 1st child | Ref. | | Ref. | |
| | 2nd or 3rd child | 1.0 | (0.8–1.4) | 1.1 | (0.8–1.5) |
| | 4th child or higher | 0.5*** | (0.4–0.7) | 0.7* | (0.5–1.0) |
| **Educational** | | | | | |
| | No education | Ref. | | Ref. | |
| | Primary | 2.2*** | (1.5–3.1) | **1.8**[**] | **(1.2–2.7)** |
| | Secondary | 3.7*** | (2.6–5.3) | **2.3**[***] | **(1.5–3.5)** |
| | Higher | 8.9*** | (4.3–18.3) | **3.1**[*] | **(1.2–7.7)** |
| **Occupation (N = 3,100)** | | | | | |
| | Not working | Ref. | | Ref. | |
| | Professional | 3.8[***] | (2.0–7.0) | 1.4 | (0.7–2.8) |
| | Sales | 1.0 | (0.7–1.4) | 0.7 | (0.5–1.0) |
| | Agricultural | 0.7* | (0.5–1.0) | 1.0 | (0.7–1.4) |
| | Services | 4.9[**] | (1.8–13.5) | 2.6 | (0.9–7.2) |
| | Manual labor | 1.7** | (1.2–2.3) | 1.3 | (0.9–1.9) |
| **Wealth index** | | | | | |
| | Poorest | Ref. | | Ref. | |
| | Poorer | 2.4*** | (1.7–3.2) | **1.7**[**] | **(1.2–2.3)** |
| | Middle | 2.2*** | (1.6–3.1) | **1.5**[*] | **(1.1–2.2)** |
| | Richer | 3.0*** | (2.1–4.2) | **1.9**[**] | **(1.2–2.8)** |
| | Richest | 6.4*** | (3.4–12.0) | **3.2**[**] | **(1.5–6.8)** |
| **Residence** | | | | | |
| | Urban | Ref. | | Ref. | |
| | Rural | 0.4*** | (0.3–0.6) | 0.8 | (0.5–1.1) |
| **Region** | | | | | |
| | Plain | Ref. | | Ref. | |
| | Tonle Sap | 0.8 | (0.6–1.1) | 1.2 | (0.9–1.7) |
| | Coastal | 1.0 | (0.6–1.5) | 1.3 | (0.8–2.1) |

(*Continued*)

**Table 3.** (Continued)

| Variables | | Four or more ANC visits | | | |
|---|---|---|---|---|---|
| | | Unadjusted (N = 3,162) | | Adjusted (N = 3,100) | |
| | | OR | 95% CI | AOR | 95% CI |
| | Plateau/Mountain | 0.3*** | (0.2–0.4) | 0.5*** | (0.3–0.6) |

**Ref** = reference value.

\* $p < 0.05$

\*\* $p < 0.01$

\*\*\* $p < 0.001$.

**Notes:** *Survey weights are applied to obtain weighted percentages.* \***Plains:** *Phnom Penh, Kampong Cham, Tbong Khmum, Kandal, Prey Veng, Svay Rieng, and Takeo;* **Tonle Sap:** *Banteay Meanchey, Kampong Chhnang, Kampong Thom, Pursat, Siem Reap, Battambang, Pailin, and Otdar Meanchey;* **Coastal/sea:** *Kampot, Kep, Preah Sihanouk, and Koh Kong;* **Mountains:** *Kampong Speu, Kratie, Preah Vihear, Stung Treng, Mondul Kiri, and Ratanak Kiri.*

insurance coverage were more likely to attend four or more ANC visits (AOR = 1.6, 95% CI: 1.1–2.4). Women with higher education (AOR = 3.1, 95% CI: 1.2–7.7), secondary education (AOR = 2.3, 95% CI: 1.5–3.5), and primary education (AOR = 1.7, 95% CI: 1.2–2.7) were more likely to have four or more ANC visits than women without any formal education. The odds of having four or more ANC visits were more significant for women from the wealthiest households than for those from the poorest households: richest households (AOR = 3.2; 95% CI: 1.5–6.8), richer households (AOR = 1.9; 95% CI: 1.2–2.8), and middle households (AOR = 1.5; 95% CI: 1.1–2.2). However, the odds of having four or more ANC visits were lower in unmarried women than in married women (AOR = 0.5; 95% CI: 0.3–0.8).

## Discussion

We analyzed the most recent 2021–2022 CDHS data to examine the relationship between health insurance coverage and receiving four or more ANC visits during pregnancy. Overall, 24.9% of women reported having health insurance coverage among women of reproductive age who gave birth within two years of the survey. This finding is slightly similar to lower-middle-income countries, where 27.3% of women had health insurance coverage [19]. This is higher than in low-income countries, where 7.9% of women have health insurance coverage [19]. However, lower than in upper-middle-income countries, 52.5% of women had health insurance coverage [19]. Since the formal launch of the Cambodia National Social Security Fund (NSSF) with the Health Insurance Scheme in 2008, the proportion of women with health insurance coverage has increased from 16% in 2014 to 22% in 2021–2022 [5]. This proportion exponentially increased due to the Royal Government of Cambodia's implementation of the NSSF for all workers in the formal and informal sectors of the economy [10]. Moreover, it has plans to extend the healthcare benefits under the NSSF to the family members of the employees as well [16].

This study found that women with health insurance coverage were 1.6 times more likely to attend four or more ANC visits during pregnancy. Previous studies documented the positive relationship between health insurance and the number of ANC visits among women of reproductive age [8,9,19]. Health insurance eliminates the financial barrier to accessing maternal health services caused by out-of-pocket payments. It has a beneficial effect in reducing the number of low-birth-weight babies born and child mortality [17,21]. The result is more equitable access to care, potentially improving maternal health outcomes [9,19]. The MoH has since raised the minimum standard for ANC visits during pregnancy to at least four trips [20,22].

The dramatically significant increase in the highest prevalence of four or more ANC visits was an effort by the Royal Government of Cambodia, which has strengthened health facilities across the country, particularly in rural areas, improved infrastructure, provided essential medical equipment and supplies, increased the number of midwives, expanded antenatal care, and provided more skilled medical practitioners at childbirth to ensure safe delivery practices. Furthermore, to encourage early and routine ANC visits, the government is offering pregnant women a monetary incentive of 20 US dollars for each visit during a maximum of four ANC visits at any health facility with a contract with the National Social Security Fund (NSSF) [14,23].

This study found that increased education and household wealth index increased the likelihood of four or more ANC visits. Women with education and greater wealth index were more likely to attend four or more ANC visits. This aligns with previous evidence around socio-economic inequalities in maternal health service utilization in Cambodia and South Asia [24,25]. Education can increase women's awareness of the importance of ANC, while higher wealth can provide the financial means and access necessary to attend at least four ANC visits [26]. Additionally, education gives women the power to decide whether to seek medical attention and enables them to recognize warning signs of pregnancy complications. Moreover, women from higher-income households were more likely to be able to cover the costs of care-seeking, including any related expenses and transportation [24,25]. Thus, in this study, women with higher education levels and household wealth indexes had the highest proportion of health insurance coverage.

This study has several strengths. First, it used the most recent women's data from the 2021–2022 CDHS, an extensive representative national population-based household survey with a high response rate of 97%. Second, the recall bias has been minimized by limiting the analysis to the women's most recent deliveries within the last two years preceding the survey [5]. Third, the complex survey design and sampling weights were incorporated into the analysis of descriptive statistics, Chi-square test, and Logistic regression model, which enabled us to generalize our findings to the population of WRA in Cambodia. In addition, DHS data were collected using validated survey methods and highly trained data collectors, contributing to improved data quality [27]. Last, to our knowledge, this is the first study to report the association between health insurance coverage and ANC visits in Cambodia. After controlling for sociodemographic factors, we found significant associations between health insurance coverage and attendance at four or more ANC visits.

Despite this, there are several limitations. First, this study used a secondary analysis, so it did not address health institution factors of antenatal care utilization and service availability; hence, this study could not explore the quality of ANC services, though the quality of healthcare services plays a vital role in patient satisfaction and use. Second, the study's cross-sectional nature could not assist in the temporal relationship of variables, including the number of years since women joined health insurance for ANC utilization. Therefore, further study should be conducted to identify factors related to health institutions. In addition, antenatal care utilization should be performed based on the new WHO guidelines revised in 2016 at the national level [22].

Moreover, longitudinal studies that address comprehensive variables should be studied. Third, we excluded other factors, such as maternal complications and women's empowerment indicators, that could affect the use of maternal care. Lastly, CDHS did not assess a direct measure of maternal health literacy.

## Conclusion

This is the first study to report the association between health insurance coverage and ANC visits in the healthcare setting in Cambodia. Cambodian pregnant women attend four or more

antenatal care visits, which is slightly high. However, it still needs to be satisfactory. Health insurance coverage among women in Cambodia is relatively low. Moreover, we found that women with health insurance, women with education, and being rich in the wealth quintile were strong predictors of women attending four or more ANC visits. There is a need to pay close attention to improving the uptake of health insurance among women of reproductive age, especially targeting women with no education, from low-income families, and women who reside in rural areas. Policymakers may need to prioritize women of reproductive age in designing and implementing health insurance programs to increase their uptake. This would provide financial risk protection, facilitate access to maternal health services, and possible attainment of Cambodia's SDG 3 targets.

## Acknowledgments

The authors would like to thank DHS-ICF, who approved the data used for this paper. And we thank Mr. Sopheap Suong, Flinders University, Adelaide, South Australia, Australia, who provided helpful, professional proofreading.

## Author Contributions

**Conceptualization:** Samnang Um, Channnarong Phan, Khun Veha, Soklim Pay, Darapheak Chau.

**Data curation:** Samnang Um, Khun Veha, Soklim Pay.

**Formal analysis:** Samnang Um, Darapheak Chau.

**Investigation:** Samnang Um.

**Methodology:** Samnang Um.

**Project administration:** Samnang Um, Channnarong Phan, Leng Dany, Khun Veha.

**Resources:** Leng Dany.

**Supervision:** Samnang Um, Leng Dany.

**Validation:** Samnang Um, Darapheak Chau.

**Visualization:** Samnang Um.

**Writing – original draft:** Samnang Um, Leng Dany, Khun Veha, Soklim Pay.

**Writing – review & editing:** Samnang Um, Channnarong Phan, Darapheak Chau.

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
