## [Decision Letter · Decision Letter 0]

20 Jun 2024

PGPH-D-24-00222

Health insurance coverage and antenatal care utilization in Cambodia: Analysis of Cambodia Demographic and Health Survey 2021-22

Dear Dr. Um,

Thank you for submitting your manuscript to PLOS Global Public Health. After careful consideration, we feel that it has merit but does not fully meet PLOS Global Public Health’s publication criteria as it currently stands. Therefore, we invite you to submit a revised version of the manuscript that addresses the points raised during the review process.

We look forward to receiving your revised manuscript.

Kind regards,

Abdur Razzaque Sarker, PhD

Academic Editor

Journal Requirements:

Additional Editor Comments (if provided):

This paper tried to examine the effects of health insurance coverage and antenatal care (ANC) utilization among pregnant women in Cambodia. Although the reviewers raised some concerns regarding this paper. My major concern is the analysis regarding the table 4 multiple logistic regression model where the authors showed a significant relationship among the insured and non-insured mother regarding the recommended ANC service utilization. The authors conclude compared to women without health insurance, those with health insurance coverage were 1.6 times more likely to attend four or more ANC visits. However, Table 4 also indicated that, most of the explanatory variables have a positive relationship with outcome variables. Even we observed that richest and higher educated mother utilized 3.2 and 3.1 times higher than their counterpart which is higher than having insurance variable. Therefore, I wondered education and wealth more important that having health insurances. Please clarify? The paper is fine, if the authors change the title of this paper to find out the determinants of ANC service utilization using Cambodia DHS data. However, to examine the effect of health insurance, I would like to see the reanalysis with two separate models for the mothers who were belonged to the health insurance coverage and who have not. Indeed, without propensity score matching, I believed the impact of health insurance on ANC service utilization may be questionable. See the related papers

https://doi.org/10.1136/bmjopen-2020-040062

https://doi.org/10.1093/heapol/czw135

Reviewers' comments:

Reviewer's Responses to Questions

**Comments to the Author**

1. Does this manuscript meet PLOS Global Public Health’s publication criteria? Is the manuscript technically sound, and do the data support the conclusions? The manuscript must describe methodologically and ethically rigorous research with conclusions that are appropriately drawn based on the data presented.

Reviewer #1: Yes

Reviewer #2: Yes

Reviewer #3: Yes

Reviewer #4: Yes

Reviewer #5: Yes

2. Has the statistical analysis been performed appropriately and rigorously?

Reviewer #1: Yes

Reviewer #2: Yes

Reviewer #3: Yes

Reviewer #4: Yes

Reviewer #5: Yes

3. Have the authors made all data underlying the findings in their manuscript fully available (please refer to the Data Availability Statement at the start of the manuscript PDF file)?

Reviewer #1: Yes

Reviewer #2: Yes

Reviewer #3: No

Reviewer #4: Yes

Reviewer #5: Yes

4. Is the manuscript presented in an intelligible fashion and written in standard English?

Reviewer #1: Yes

Reviewer #2: Yes

Reviewer #3: Yes

Reviewer #4: No

Reviewer #5: No

5. Review Comments to the Author

Reviewer #1: please correct 15-59 years as 15-49 years in line number 116 (page 4). Complete the line 133 -descriptive and analytical analysis or something like this. Make the line 141 correct, there is no verb in this sentence. Rewrite the sentence /line number149. Correct/rewrite line number 257,258. Read carefully the full text .

Reviewer #2: Thank you for sharing the manuscript and would like to congratulate them for their nice work. My few comments/clarification questions are incorporated in the manuscript.

1. Did the study assess the cut-off period to identify the association between access to maternal health and years of membership/participation in health insurance?

2. What is the particular interest of the researchers only using 2021-2022 data?

3. How do you assess the confounding effects of other variables such as income, level of education? Because those with better education and income are likely to attend 4 or more ANC visits.

4. Just for curiosity, is those women who did not attend any ANC visit categorized under less than 4 ANC visit?

5. Did the study assess the association between the number of years since they joined the insurance and ANC use?

Reviewer #3: Thank you for the opportunity to review this manuscript titled "Health insurance coverage and antenatal care utilization in Cambodia: Analysis of Cambodia Demographic and Health Survey 2021-22"

Title: I can see that title has two outcome variables (Health insurance coverage and ANC utilization) and no independent variable. It would be better to have this put to look a complete topic. Example; you can write, "Factors Associated with Health Insurance and ANC Utilization..........."

Methods: In the outcome variable section you stated that "This study's outcome was the number of ANC visits..." I can see that you left out another important outcome variable, i.e Health Insurance Coverage. Please update this section. Further, I can see that you used these two outcome variables very well in your result section (see Table 2 & Table 3).

Results: Update your topic that reflect the independent variables used in result. I gave a suggestion how the title can be improved.

Data sharing: Write a statement on data related to your study and indicate the URL link for the data

Reviewer #4: The manuscript adequately responds to the research question based on the predefined scope. This is a secondary analysis of data from a demographic and health survey. The author should be keen not to describe the survey in the methodology instead of the approaches of the current study. Minor comments are included in the attached manuscript for author review.

Reviewer #5: I am glad to have had the opportunity to review this pertinent and interesting paper. The paper addresses a relevant issue, both socially and scientifically; however, I believe some changes could clarify and improve the current manuscript.

Please find my comments below:

**Major Comment

• In methodology clarification of the independent variables could be provided using proper references to justify the categorization of the variables.

• Regarding results, in general, the section is clear. The discussion and conclusion also look good.

**Minor Comment

• There are some typos that must be addressed with proper care and also some sentences lack coherence. These issues should be taken care of.

6. PLOS authors have the option to publish the peer review history of their article (what does this mean?). If published, this will include your full peer review and any attached files.

**Do you want your identity to be public for this peer review?** For information about this choice, including consent withdrawal, please see our Privacy Policy.

Reviewer #1: **Yes: **Israt Jahan Kakoly

Reviewer #2: **Yes: **Alebel Yaregal Desale

Reviewer #3: No

Reviewer #4: No

Reviewer #5: No

---

## [Decision Letter · Decision Letter 1]

21 Aug 2024

PGPH-D-24-00222R1

The effect of health insurance coverage on antenatal care utilization in Cambodia: A secondary analysis of Cambodia Demographic and Health Survey 2021-2022

Dear Dr. Um,

Thank you for submitting your manuscript to PLOS Global Public Health. After careful consideration, we feel that it has merit but does not fully meet PLOS Global Public Health’s publication criteria as it currently stands. Therefore, we invite you to submit a revised version of the manuscript that addresses the points raised during the review process.

We look forward to receiving your revised manuscript.

Kind regards,

Abdur Razzaque Sarker, PhD

Academic Editor

Journal Requirements:

Additional Editor Comments (if provided):

There is a great deal of room for improvement in both writing and grammar. I advise the authors to work with a writing coach or copy editor to improve the flow and readability of the text.

Reviewers' comments:

Reviewer's Responses to Questions

**Comments to the Author**

1. If the authors have adequately addressed your comments raised in a previous round of review and you feel that this manuscript is now acceptable for publication, you may indicate that here to bypass the “Comments to the Author” section, enter your conflict of interest statement in the “Confidential to Editor” section, and submit your "Accept" recommendation.

Reviewer #3: All comments have been addressed

Reviewer #4: (No Response)

2. Does this manuscript meet PLOS Global Public Health’s publication criteria? Is the manuscript technically sound, and do the data support the conclusions? The manuscript must describe methodologically and ethically rigorous research with conclusions that are appropriately drawn based on the data presented.

Reviewer #3: Yes

Reviewer #4: Partly

3. Has the statistical analysis been performed appropriately and rigorously?

Reviewer #3: Yes

Reviewer #4: Yes

4. Have the authors made all data underlying the findings in their manuscript fully available (please refer to the Data Availability Statement at the start of the manuscript PDF file)?

Reviewer #3: Yes

Reviewer #4: Yes

5. Is the manuscript presented in an intelligible fashion and written in standard English?

Reviewer #3: Yes

Reviewer #4: No

6. Review Comments to the Author

Reviewer #3: Thank you for addressing all the comments

Reviewer #4: The authors have fairly attempted to address the previous comments. However, the manuscript still needs a thorough proof reading. More suggestions can be found in the attachment.

7. PLOS authors have the option to publish the peer review history of their article (what does this mean?). If published, this will include your full peer review and any attached files.

**Do you want your identity to be public for this peer review?** For information about this choice, including consent withdrawal, please see our Privacy Policy.

Reviewer #3: No

Reviewer #4: No

---

## [Editor Report · Decision Letter 2]

27 Sep 2024

PGPH-D-24-00222R2

The effect of health insurance coverage on antenatal care utilization in Cambodia: A secondary analysis of Cambodia Demographic and Health Survey 2021-2022

Dear Dr. Um,

Thank you for submitting your manuscript to PLOS Global Public Health. After careful consideration, we feel that it has merit but does not fully meet PLOS Global Public Health’s publication criteria as it currently stands. Therefore, we invite you to submit a revised version of the manuscript that addresses the points raised during the review process.

We look forward to receiving your revised manuscript.

Kind regards,

Abdur Razzaque Sarker

Academic Editor

Journal Requirements:

Additional Editor Comments (if provided):

There are still some typo errors. I would like to suggest a thorough copy edit of the revised version.
---

## [Editor Report · Decision Letter 3]

18 Oct 2024

The effect of health insurance coverage on antenatal care utilization in Cambodia: A secondary analysis of Cambodia Demographic and Health Survey 2021-2022

PGPH-D-24-00222R3

Dear Dr. Um,

We are pleased to inform you that your manuscript 'The effect of health insurance coverage on antenatal care utilization in Cambodia: A secondary analysis of Cambodia Demographic and Health Survey 2021-2022' has been provisionally accepted for publication in PLOS Global Public Health.

Best regards,

Abdur Razzaque Sarker, PhD

Academic Editor